# OpenReview forum: "Redistributing Token-Level Rewards from Sequence-Level Feedback"
_TMLR — Rejected by TMLR_

### Review · Reviewer_X43t · 2025-02-10

**Summary Of Contributions:**

The paper studies an interesting problem of how sequence-level reward models can be used to provide token-level feedback and also its implications on preference optimization. To extract token-level reward for a given sequence, the authors proposed using the incremental reward contribution from each token. The proposed reward distribution approach, RED, improves performance (GPT4 win-rate and average reward) of PPO on three standard benchmark datasets.

**Audience:**

Yes

**Broader Impact Concerns:**

No ethical concerns

**Claims And Evidence:**

No

**Requested Changes:**

1. For contribution 2 “Low Computational Costs” -- I think this is an overclaim, since the paper does not reduce the time complexity of PPO. A better way of describing it may be “RED does not lead to Additional Overhead”.


2. Although for the example in Figure 1, the idea of token-level reward makes sense, it is difficult to comprehend how the token-level rewards are distributed for coding/math instruction prompts. The authors should also add example for these kind of prompts.


3. In Section 4.6, the authors plot the rewards vs step and KL vs step for different baselines. It would be really interesting to visualize the rewards vs KL and GPT4 win-rate vs KL plot of different baseline approaches.


4. [Minor] For comprehensiveness of the evaluations, I shall recommend the authors also report the coherence and diversity of the generated responses (as defined in [1]) using RED compared to the base model.


5. There are some typos in the draft. For example in the “Base model and Benchmark” section, it should be LLaMA-7B and LLaMA-3-8B. Please correct them in the revised draft.

[1] Khanov, M., Burapacheep, J. and Li, Y., ARGS: Alignment as Reward-Guided Search. In The Twelfth International Conference on Learning Representations.

**Strengths And Weaknesses:**

>Strengths

1. The overall paper presentation is good, and the research problem of extracting token-level rewards from sequence-level reward models has been well-articulated to the reader.

2. On three standard benchmark datasets (Nectar, TLDR, and PKU-SafeRLHF), RED shows improvement in the performance of the PPO model as compared to other baselines.

>Weaknesses

1. Although the empirical results in the paper clearly show that RED helps in improving performance of PPO, the exact reason for this improvement is not clear. The authors in contribution 1 claim that: “By providing token-level rewards, …. enhances learning by offering immediate and relevant information. The …. delayed rewards that may be less informative”. However, why sequence-level rewards may be less informative and how token-wise rewards can be helpful is not clear from the current draft.

2. There may be cases where due to tokenization, a single word may be divided into multiple tokens, which may not have any semantical meaning. In a simple example using LLAMA tokenizer, the word “harmless” is tokenized as “h”, “arm”, “less”.  In this case, how distributing the rewards would be helpful is not clear. This problem can become much more complex for math/coding instructions.

3. In Section 4.6, the KL vs step shows that PPO-RED results in a significant increase in KL divergence compared to PPO and PPO-ABC. A larger KL would intuitively indicate that the model has probably lost the capabilities learned during SFT. It would be better to visualize the rewards vs KL and GPT4 win-rate vs KL plots.

4. [Minor] The related works section could have been more extensive.

---

> ### Author Response · Authors · 2025-02-28
>
> Thank you for your thorough review and thoughtful comments. We have provided detailed responses to each of your queries below.
>
> ****Q1: How token-wise rewards can be helpful is not clear.****
>
> A1: In RLHF, each token corresponds to a time-step action in the RL framework. Sequence-wise rewards, which only provide guidance at the final token of a generated sequence, lead to a sparse reward scenario, making it challenging for the language model to optimize. This is because the model struggles to discern which specific tokens contribute to the overall success or failure of the sequence. In contrast, token-wise rewards address this issue by providing fine-grained feedback at each time step, enabling more efficient learning. The limitations of sequence-wise guidance and the advantages of more granular reward signals are also well-discussed in the literature [1–3].
>
> For example, consider the prompt "Who was the first king of Belgium?" and the corresponding long generation: "The first king of Belgium was Leopold ...... <EOS>." In this scenario, the reward model assigns a positive evaluation score to the entire generated sequence. Clearly, the entity "Leopold" contributes most significantly to the success of the response. However, in traditional RLHF, the rewards for each token preceding the "<EOS>" token are set to zero.
>
> In contrast, the reward redistribution method allocates rewards based on the individual contributions, or "credits" of each token. This approach allows tokens like "Leopold" to receive higher rewards for their pivotal role in correctly answering the prompt. By doing so, the method facilitates more effective learning (i.e, with "more accurate training signals") for the LLM, enabling it to identify and reinforce the patterns that lead to accurate and meaningful responses.
>
> ****Q2: Cases where, due to tokenization, a single word may be divided into multiple tokens.****
>
> A2: Such cases are relatively rare with advanced tokenization schemes used by state-of-the-art language models. For instance, the tokenizer for LLaMA includes over 32,000 tokens. Moreover, while occasional inaccuracies in tokenization might arise, they generally do not significantly impact the overall semantic representation of a sentence.
>
> As for specialized domains such as mathematics or coding, we cannot guarantee that our method adequately addresses this issue. Please refer to Q3 for further details.
>
> ****Q3: Behaviour in math/coding instructions.****
>
> A3: There are some questions related to math/code in the Nectar dataset, we have extracted them to make some analysis.
> However, the results show that the base model performs poorly in math/code tasks and usually generates wrong results. Moreover, the  SFT model, RL model, and our method can not improve the model performance of these tasks.
> We think the reasons are as follows:
>
> (1) The model in our experiments is 7B/8B, and behaves poorly in such tasks.  More large models with the size of 80B may be capable of math/code tasks, but our computational resources do not support the optimization of 80B models.
>
> (2) Performance improvement on math/code requires a large amount of training data and multistage training[4].
>
> (3) The math/code scenario usually requires multi-step reasoning, where using a high-quality PRM[5,6] is usually an efficient way, but our methods are not particularly designed for improving the reward model itself and providing signals for reasoning steps.
>
> In conclusion, we can not quantify whether and how our method can improve the math/code ability of LLMs, and we apologize for this. We will leave this for future work.
>
> ****Q4: Visualizing the rewards of GPT-4 win rate vs. KL plots.****
>
> A4: Thank you for the suggestion. In the revised manuscript, we have added Figure 3(c)(d) to visualize the relationship between the GPT-4 win rate and KL divergence, as requested.
>
> ****Q5: Related works****
>
> A5: Thank you for your comments, and have incorporated additional references in the related work section of the revised manuscript, including GRPO[4] and inverse-Q[2].

---

> ### Author Response · Authors · 2025-02-28
>
> ****Q6: Overclaim of lower computational costs.****
>
> A6: Thank you for bringing this to our attention. We have revised the manuscript to provide a more accurate presentation of the computational cost of our method.
>
>
> ****Q7: Coherence and diversity [7].****
>
> A7: We appreciate the suggestion to evaluate coherence and diversity. We have computed these metrics and included the results in the appendix of the revised manuscript. Our findings indicate that all optimized LLMs demonstrate similar performance in both coherence and diversity.
>
> Additionally, we did not rely on traditional automatic evaluation metrics (e.g., BLEU, ROUGE, METEOR) in the main body of the paper. This decision stems from the primary objective of RLHF, which is to align models with human preferences. As previous studies [8] have shown, these traditional metrics often correlate poorly with human judgments, making them less suitable for our evaluation framework.
>
> ****Q8: Typos.****
>
> A8: Thank you for pointing out the typographical errors. We have thoroughly reviewed the manuscript and corrected all identified errors in the revised version to improve clarity and readability.
>
> [1] Dense reward for free in reinforcement learning from human feedback. ICML2024
>
> [2] Token Level Reinforcement Learning for Aligning Large Language Models without Preference Data. EMNLP2024
>
> [3] DPO Meets PPO: Reinforced Token Optimization for RLHF. arXiv preprint arXiv:2404.18922
>
> [4] DeepSeekMath: Pushing the Limits of Mathematical
> Reasoning in Open Language Models https://arxiv.org/pdf/2402.03300
>
> [5] Weak-to-strong generalization: Eliciting strong capabilities with weak supervision. ICML2024
>
> [6] Math-shepherd: Verify and reinforce llms step-by-step without human annotations. ACL2024
>
> [7] ARGS: Alignment as reward-guided search. ICLR2024
>
> [8] Direct preference optimization: Your language model is secretly a reward model. NeurIPS2023

---

### Review · Reviewer_auZF · 2025-02-11

**Summary Of Contributions:**

In the context of LLM fine-tuning using reinforcement learning (RL), this paper proposes to use a trained reward model as potential-based reward shaping term to obtain dense reward signals.

**Audience:**

No

**Broader Impact Concerns:**

None.

**Claims And Evidence:**

No

**Requested Changes:**

The paper should be:
- reorganized to avoid repetition and improve presentation.
- checked for typos.
If my understanding of SMDP is correct, the authors should remove any reference to it.

The authors should also provide the results for a fixed \beta_c without hyperparameter tuning (which would constitute a realistic setting).

**Strengths And Weaknesses:**

STRENGTHS

The problem tackled by this paper is important. An efficient technique could accelerate LLM fine-tuning, which would be especially impactful in AI alignment. The proposed technique is simple and relies on well-understood RL technique (i.e., potential-ased reward shaping).

WEAKNESSES

From the RL viewpoint, the proposed method has limited innovation. In addition, to obtain the best performance in a given task, it seems that hyperparameter tuning of \beta_c is needed (as recognized by the authors themselves in the appendix), which makes the approach impractical.

The current version of the paper needs some important revision:
- I think the motivation for the work is misleading and incorrect. The trained reward model actually provides a reward value on the whole generated sequence, not only on the EOS token.
- The authors refer several times to sequence-Markov decision processes (SMDP), however I believe that this model does not apply to LLM fine-tuning. As explained in Section 2.1, a state contains both the prompt and the sequence generated so far. A reward function that takes a final state (which would include the whole generated sequence) would be Markov.  In their presentation, the authors seem to confuse token, state, and sequence.
	In addition, since the returns after reward shaping are changed, it's not clear how the results from SMDPs applies.
- The paper needs some reorganization and rewriting. Some explanation happens too late, e.g.,
	The explanation of notations and symbols should happen the first time they appear.
	The discussion of related work is repeated.
	The explanation of the meaning of RS and LAG should appear in the beginning of Section 4.
- The mathematical formalization needs to be reviewed, e.g.,
	some less than sign should be less or equal instead (e.g., Section 2.1),
	why a discount factor strictly smaller than 1 is introduced, but then never used?
	\tilde r_t in the bottom of page 5 seems to be a typo.
	\mathcal R_cost vs \mathcal C_\phi should be explained.
	The notations in the appendix should be consistent with the main paper (e.g., \phi).
- The paper should be checked for typos (e.g. Section 4.1). Some sentences are repeated in the appendix (above App. B). Tables 5-7 are figures.
- The authors should discuss how the hyperparameters were chosen for all the methods.
In Fig. 4, are the rewards comparable for the different \beta_c, since the reward definition changes?

---

> ### Author Response · Authors · 2025-02-28
>
> We sincerely appreciate your thorough review and invaluable feedback. Below, we provide detailed responses to all your comments or concerns.
>
> ****Q1: Motivation****
>
> A1: Thank you for your concern. In the current reward model, an evaluation score at "<EOS>"  is provided for the entire generation sequence. Such rewards are both sparse and delayed, making it challenging for RL algorithms to effectively optimize the policy. This issue has been identified and explored in previous works [1–3].
>
> For example, consider the prompt "Who was the first king of Belgium?" and the corresponding long generation: "The first king of Belgium was Leopold ...... <EOS>." In this scenario, the reward model assigns a positive evaluation score to the entire generated sequence. However, in traditional RLHF, the rewards for each token before the "<EOS>"  token are set to zero.
>
> This situation motivates us to redistribute the rewards based on the credit of each individual token. In the example, the token "Leopold" receives a relatively high reward since it accurately answers the prompt. This facilitates the LLM in learning the correct response patterns more efficiently.
>
> ****Q2: How do the results from SDPs apply?****
>
> A2: We apologize for any confusion. The definitions of MDPs and SDPs are provided in Section 2.1.
> We maintain that our method adheres to the SDPs because the redistributed rewards in our approach do not satisfy the Markov property.
>
> The Markov Property stipulates that the future state of a system is conditionally independent of its past states given the present state. Formally, for any timestep $t$, the probability of transitioning to the next state depends solely on the current state, and not on the sequence of states that preceded it.
>
> In our method, the state at timestep $t$ is defined as $(x,y_{<t})$, and the generated action is $y_{t}$.
> As described in Eq. (6), the reward at each timestep $t$ depends not only on the current state $(x,y_{< t})$ and action $y_t$ but also on the previous state $(x,y_{<t-1})$ and action $y_{t-1}$. This dependency violates the Markov Property because the future rewards are influenced by the history of states, not just the current state. Despite this violation, the summation of our redistributed rewards remains equal to the original return.
>
> Therefore, our method operates within the framework of SDPs by maintaining the equivalence of total rewards while allowing for dependencies that extend beyond the Markov Property. We have added further explanations to clarify this point in the revised manuscript.
>
> ****Q3: Mathematical formalization.****
>
> A3: Thank you for highlighting the need for clearer mathematical formalization.
> The discount factor  $\gamma$ is typically set to a value strictly less than 1 to ensure the convergence of the RL algorithm. In our experiments, we have set  $\gamma=0.99$, as detailed in the Appendix.
>
> Meanwhile, we have thoroughly revised the manuscript to enhance the clarity and consistency of our notations.
>
> 1. Notation Clarification: We have provided additional explanations for all notations used throughout the paper.
>
> 2. Consistency Between Sections: We have harmonized the notations in the appendix with those in the main text.
>
> These revisions aim to make the mathematical sections more accessible and accurate, addressing the concerns raised.

---

> ### Author Response · Authors · 2025-02-28
>
> ****Q4: Hyperparameters chosen. Are the rewards comparable in Fig. 4?****
>
> A4: We appreciate your question.
>
> ****Hyperparameter Selection****: Typically, setting
> $\beta_c=1$  yields strong performance across various tasks. However, in certain scenarios, selecting an appropriate value for
> $\beta_c$ can enhance training stability and achieve superior performance. We have discussed this in Section 3.2 and have emphasized it further in the revised manuscript. In our experiments,  $\beta_c=1$  was used in most scenarios, except for the TL;DR dataset with LLaMA3, where $\beta_c=0.5$.
>
> ****Rationale for Adjusting $\beta_c$:**** The TL;DR task involves generating summaries where the '"<EOS>" token plays a crucial role in model generation. Adjusting $\beta_c$  in this context makes the reward redistribution more effective, addressing the task's specific demands.
>
> ****Comparability of Rewards in Fig. 4:**** Following the approach in [1] , the reward without KL regularization (as defined in Eq. (8)) is a convex combination of token-wise and sequence-wise rewards. Here,
> $\beta_c$
> controls the trade-off between these two components. Importantly, rewards generated with different
> $\beta_c$  values belong to the same equivalence class, meaning they are comparable despite changes in reward definitions. Therefore, the rewards depicted in Fig. 4 remain comparable across different
> $\beta_c$  settings.
>
> ****Q5: Paper reorganization.****
>
> A5: Thank you for your constructive feedback. We have reorganized the paper according to your comments and corrected the typographical errors.
>
> [1] Dense reward for free in reinforcement learning from human feedback. ICML2024
>
> [2] Weak-to-strong generalization: Eliciting strong capabilities with weak supervision. ICML2024
>
> [3]  ARGS: Alignment as reward-guided search.ICLR2024

---

### Review · Reviewer_vMky · 2025-02-15

**Summary Of Contributions:**

Summary: This paper develops a reward-shaping approach to improve the process of RLHF. The approach consists of a heuristic metric for allocating rewards token-by-token in a sequence rather than only rewarding the last token. The approach is tested using LLaMA on multiple datasets with ablations to show that the approach, REward reDistribution (RED), improves the performance of PPO and other baseline RL methods.

**Audience:**

Yes

**Claims And Evidence:**

No

**Requested Changes:**

Please address all weaknesses noted above.

**Strengths And Weaknesses:**

Strengths:

The paper is generally easy to read and presents an intuitive method for addressing the credit assignment problem in fine-tuning LLMs through reinforcement learning.
The equations are clear.
The results are thorough with clear ablations.
The tables with examples are quite helpful.
Weaknesses:

Table 1 only shows LLaMA and LLaMA3 base models, likely due to it being open source. GPT-4 was used as the "judge." It would be nice to see what the results would look like with LLaMA judging LLaMA.

Pseudocode with line-by-line walk-throughs would be helpful to enhance reproducibility. The code may be relatively simple but would nonetheless be helpful.

Section 3.3 is quite important and states major assumptions and properties. While some of it may seem intuitive on a surface level, it would be helpful to give a formal proof for these properties / analysis. A working example with visualizations would also be helpful to walk the reader through the intuition for this analysis. The presented analysis is not rigorous enough for publication in its current form.

The paper is missing a rigorous empirical "proof" that convinces the reviewer that the credit assignment problem is being properly solved. Figure 1 shows a distribution of rewards being more accurate (according to some nebulous definition of accurate -- perhaps envisaged by Equation 10. However, there is no validation to show this is the best or correct specification. The results do show that this specification is helpful, but the results do not show it is right. I would recommend the following:

Perform a sensitivity analysis to show how well the model performs on its primary metric (e.g., correctly answering questions) vs. how accurate the rewards are distributed.
A human eval would be helpful to compare how accurate the rewards are predicted by the paper's metric vs. human-assigned rewards. Training on those human rewards would be a helpful baseline to include.
Having to tune \Beta and \Beta_c is expected but not ideal.
-"shwon" should be "shown' in the first line of the second paragraph of Section 3.2

---

> ### Author Response · Authors · 2025-02-28
>
> Thank you for your careful review and insightful comments. We have addressed each of your queries in detail below.
>
> ****Q1: LLaMA judging LLaMA****
>
> A1: We appreciate your constructive feedback. In our initial experiments, we employed LLaMA-7B and LLaMA3-8B as evaluators to compare responses from our SFT and RLHF-enhanced models. Interestingly, the LLaMA-based judges exhibited a consistent preference for the SFT model over the RLHF counterpart across all datasets.
>
> We hypothesize two primary reasons for this phenomenon:
>
> ****1. Model Capacity Constraints****: The 7B/8B-parameter LLaMA/LLaMA3 models may lack the capacity/sophistication required to reliably approximate human judgment. Their limited scale could hinder nuanced evaluation, particularly for complex alignment-focused outputs from RLHF.
>
> ****2. Intrinsic Training Bias****: As LLaMA-family models may inherently favor text distributions resembling their own training data, they systematically prefer SFT-generated outputs that align more closely with their native response patterns than with RLHF-optimized responses.
>
> Given these findings and observations, we recognize that LLaMA-based evaluation introduces significant bias risks, potentially conflating model-specific preferences with objective quality assessment Therefore, we have opted to exclude these results from the revised manuscript to maintain the integrity of our evaluations.
>
> ****Q2: Proof for properties in Sec 3.3****
>
> A2: Thank you for your insightful comments. Our method exhibits strong properties and demonstrates clear connections with the works [1, 2, 3]. We acknowledge the importance of providing rigorous proofs to support our claims. However, the relevant properties have already been thoroughly proved in these references [1, 2, 3]. Since our work does not extend the theoretical contributions beyond what is covered in these sources, we have opted not to include redundant proofs in the main manuscript. Nonetheless, we have ****provided the proof for the "unchanged optimal policy"**** in the Appendix A.2.
>
> ****Q3: Sensitivity Analysis****
>
> A3:  Quantifying the accuracy of token-wise rewards is challenging. However, to evaluate how redistributed rewards impact the performance of LLMs, we conducted a sensitivity analysis as suggested in your comment. This analysis verifies the effectiveness and robustness of our approach.
>
> We use RED as the baseline, considering it to represent the rewards with the highest accuracy.
> Meanwhile, we introduced random noise to each token of the generated sentence while keeping the overall return unchanged by adjusting the score at the final time step. Formally, we added a random noise term $\alpha \cdot r^{\text{noise}}_t$  to each $\tilde{r}_t$ for $0 \leq t \leq T-1$ and subtracted $\alpha*\sum^{T-1}_0 {r^{noise}_t}$ from  $\tilde{r}_T$, where $\alpha$ controls the intensity of the reward redistribution inaccuracy.
>
> The results of this analysis are presented in Figure 5(b) of the revised manuscript. The findings demonstrate that even with the introduction of inaccurate reward redistribution, our method consistently outperforms RLHF. This supports our claim that "even a non-optimal redistribution method can lead to desirable learning outcomes."
>
> ****Q4: Human Evaluation****
>
> A4: Thank you for highlighting the importance of human evaluation. We have conducted human assessments for each dataset, and the results are detailed in Section 4.5 of the revised manuscript. These evaluations provide qualitative insights that complement our quantitative findings, reinforcing the effectiveness of our proposed approach from a human-centric perspective.
>
> ****Q5: pseudocode****
>
> A5:  Thank you for your valuable comment. We have included the pseudocode in Appendix B.2 of the revised manuscript. Additionally, we will open-source our code to ensure reproducibility.
>
> [1] Direct preference optimization: Your language model is secretly a reward model. NeurIPS2024
>
> [2] Potential-based shaping and q-value initialization are equivalent. JAIR2011
>
> [3] Rudder: Return decomposition for delayed rewards. NeurIPS2019

---

> > ### Comment · Reviewer_vMky · 2025-03-08
> > **Response**
> >
> > This reviewer thanks the authors for their revisions as most of this reviewer's concerns have been addressed. The additional experiments, human evaluations, and inclusion of pseudocode meaningfully improve the manuscript. That said, I have a couple of remaining concerns:
> >
> > Proofs for Properties in Section 3.3 – This reviewer appreciates the reference to prior work, but the original concern was less about redundancy and more about the lack of a clear, rigorous derivation of these properties in the context of your approach. While this reviewer sees that Appendix A.2 includes a proof for the "unchanged optimal policy," a more thorough analysis of the key assumptions and their implications would strengthen the paper. Unfortunately, while the citations are helpful, these citations do not establish that these properties hold in the author's setting without a direct derivation.
> >
> > Validation of Reward Redistribution Accuracy – While the sensitivity analysis in Figure 5(b) is useful, it does not directly address whether the proposed reward redistribution is actually correct. Instead, it shows that the method is robust to noise. This reviewer's original request was for an empirical validation that the redistributed rewards align with ground-truth rewards. The human evaluations you mention in Section 4.5 are a step in this direction (if one is going for a 'human eval' standard evaluation), but it is unclear if they specifically compare human-assigned token-level rewards to those generated by your method. Clarifying this would help address the concern.

---

> > > ### Author Response · Authors · 2025-03-21
> > >
> > > Thank you for your concerns; we hope to address them with the following responses.
> > >
> > > ****Q1: A more thorough analysis of the key assumptions and their implications.****
> > >
> > > A1: Thank you for your suggestion.
> > > We have 4 important properties.
> > >
> > > (1) Unchanged Optimal Policy. We have provided a proof for this property without any specific assumptions. It is sufficient to ensure that the reward redistribution method uses the same prompt as the traditional RLHF method, which has been successfully implemented.
> > >
> > > (2) Dynamic Reward Initialization. Our method naturally accommodates both optimistic and pessimistic initializations through the term $\tilde{r}_{-1}^{RM}$, which is standard practice in reinforcement learning.
> > >
> > > (3) Convergence guarantee.  We can demonstrate that, under ****standard stochastic approximation assumptions (including Lipschitz continuity, martingale difference noise, appropriate step-size conditions, and stability of iterates)****, our method guarantees convergence to the desired attractors in a two-timescale stochastic approximation system with controlled Markov processes.
> > >
> > >
> > >
> > >
> > > a. Lipschitz Continuity: This is a common assumption for deep learning algorithms which means the behavior of the function is relatively smooth.
> > >
> > > b. Martingale Difference Noise: This assumption posits that, given past information, the expected future noise is zero, and its variance is bounded, preventing excessive fluctuations. This is a typical assumption in stochastic gradient descent and helps to ensure unbiased gradient estimates.
> > >
> > > c. Appropriate Step-Size Conditions: This assumption is also prevalent in deep learning. Specifically, the chosen learning rate $\alpha$ should satisfy the conditions
> > > $\sum_{iter=0}^{\infty} \alpha_{iter=0} = \infty$ and $\sum_{iter=0}^{\infty} \alpha_{iter=0}^2 < \infty$ to ensure algorithm convergence.
> > >
> > >
> > > d. Stability of Iterates: This assumption indicates that small disturbances will not lead to large changes in the generation process. Most deep learning algorithms achieve this through a small learning rate, while in RLHF, KL divergence and PPO algorithms facilitate this stability.
> > >
> > >
> > > We have provided the analysis of the assumptions in the revised manuscript Appendix A.
> > >
> > > (4) Robustness to Redistribution Strategy. We have provided sensitivity analysis in Appendix C.3.
> > >
> > >
> > >
> > > ****Q2: Validation of Reward Redistribution Accuracy****
> > >
> > > A2: We appreciate the feedback. Since there is no ground truth available for token-wise rewards, directly evaluating the accuracy of redistributed rewards is not feasible. While you suggest conducting a human study as an alternative, we note that this approach presents practical challenges. Specifically, generated responses often exceed 1,000 tokens, making it difficult for human annotators to assign precise labels at the token level.
> > >
> > > To address this, instead of asking annotators to label every token individually, we designed a questionnaire-based evaluation. Human annotators were asked to assess whether the redistributed rewards are overall reasonable, with a focus on verifying if the most important tokens were accurately highlighted. This approach ensures a more practical and reliable evaluation of the reward redistribution quality without imposing an overwhelming labeling burden on annotators.
> > > The results indicate that ****97\%**** of the token-wise reward sequence is generally reasonable.
> > >
> > > More details are provided in Appendix C.2 in our revised manuscript.

---

### Comment · Reviewer_vMky · 2025-02-08

Summary: This paper develops a reward-shaping approach to improve the process of RLHF. The approach consists of a heuristic metric for allocating rewards token-by-token in a sequence rather than only rewarding the last token. The approach is tested using LLaMA on multiple datasets with ablations to show that the approach, REward reDistribution (RED), improves the performance of PPO and other baseline RL methods.

Strengths:
+ The paper is generally easy to read and presents an intuitive method for addressing the credit assignment problem in fine-tuning LLMs through reinforcement learning.
+ The equations are clear.
+ The results are thorough with clear ablations.
+ The tables with examples are quite helpful.

Weaknesses:

- Table 1 only shows LLaMA and LLaMA3 base models, likely due to it being open source. GPT-4 was used as the "judge." It would be nice to see what the results would look like with LLaMA judging LLaMA.

- Pseudocode with line-by-line walk-throughs would be helpful to enhance reproducibility. The code may be relatively simple but would nonetheless be helpful.

- Section 3.3 is quite important and states major assumptions and properties. While some of it may seem intuitive on a surface level, it would be helpful to give a formal proof for these properties / analysis. A working example with visualizations would also be helpful to walk the reader through the intuition for this analysis. The presented analysis is not rigorous enough for publication in its current form.

- The paper is missing a rigorous empirical "proof" that convinces the reviewer that the credit assignment problem is being properly solved. Figure 1 shows a distribution of rewards being more accurate (according to some nebulous definition of accurate -- perhaps envisaged by Equation 10. However, there is no validation to show this is the best or correct specification. The results do show that this specification is helpful, but the results do not show it is right. I would recommend the following:
1) Perform a sensitivity analysis to show how well the model performs on its primary metric (e.g., correctly answering questions) vs. how accurate the rewards are distributed.
2) A human eval would be helpful to compare how accurate the rewards are predicted by the paper's metric vs. human-assigned rewards. Training on those human rewards would be a helpful baseline to include.

- Having to tune \Beta and \Beta_c is expected but not ideal.

-"shwon" should be "shown' in the first line of the second paragraph of Section 3.2

---

### Author Response · Authors · 2025-02-28

Thank you for your careful review and insightful comments. We have addressed each of your queries in detail below.
Additionally, we have conducted further experiments and made modifications in response to your feedback. The most significant updates are outlined as follows:

1.  We have added a proof for the important property "unchanged optimal policy" in Appendix A.2.

2. Human evaluation results have been included in Section 4.5.

3. Pseudocode has been provided in Appendix B.2.

4. We have conducted sensitivity analysis experiments, detailed in Appendix C.2.

5. Additional visualization results for our experiments are now available in Section 4.7.

6. We have added the evaluation results of diversity and consistency in Appendix B.2.

7.  We have meticulously corrected typographical errors and standardized mathematical notations throughout the manuscript to ensure clarity and consistency.

These enhancements aim to strengthen our paper and provide a more comprehensive understanding of our methodology and findings. We appreciate your valuable feedback, which has been instrumental in improving our work.

---

### Author Response · Authors · 2025-04-10

We are writing to kindly inquire about the current status of our manuscript. We appreciate the time and effort invested by the reviewers and the editorial team in evaluating our work. We want to confirm whether the reviewers have raised any additional concerns that may require clarification. If so, we would be more than happy to provide further responses or materials.

---

> ### Comment · Action_Editor_2XLg · 2025-04-11
>
> Thank you for your patience. You should expect the decision soon (1-2 weeks).
>
> Regards,
> AC

---

### Decision · Action_Editor_2XLg · 2025-04-12

**Recommendation:** Reject

**Comment:**

This paper introduces a reward redistribution technique (RED) for token-level credit assignment in RLHF, aiming to improve performance over traditional sequence-level reward models. While the paper addresses an important problem and the presentation is generally clear, the empirical and theoretical justifications remain insufficient.

The key issue is that the paper does not convincingly validate that the redistributed token-level rewards are accurate or superior to standard reward assignments. The sensitivity analysis shows robustness to noise but not correctness. Moreover, despite revisions, the lack of rigorous theoretical grounding for the method’s core assumptions (e.g., Markov properties, return equivalence) weakens the foundational claims. The method also fails to generalize to more challenging tasks such as coding and math, undermining its broader applicability.

**Audience:**

Yes

**Claims And Evidence:**

While the paper presents empirical improvements using token-level reward redistribution (RED), the core claim, particularly that RED offers more precise credit assignment and enhances learning efficiency, is not fully supported by rigorous theoretical analysis or conclusive empirical validation. The sensitivity analysis demonstrates robustness to noise but does not confirm the correctness of the redistributed rewards. Additionally, generalization to harder domains (e.g., math and code) is limited, which further weakens the claim of broad applicability.

**Resubmission Of Major Revision:**

The authors may consider submitting a major revision at a later time.